# A Study on Aeroengine Conceptual Design Considering Multi-Mission Performance Reliability

**Dalu Cao** and **Guangchen Bai** *

School of Energy and Power Engineering, Beihang University, Beijing 102206, China; cdl112358@buaa.edu.cn
* Correspondence: dlxbgc@buaa.edu.cn

**Abstract:** Owing to the realization of multi-mission adaptability requires more complex mechanical structure, the candidates of future aviation propulsion are confronted with more overall reliability problems than that of the conventional gas turbine engine. This situation is challenging to a traditional aeroengine deterministic design method. To overcome this challenge, the Reliability-based Multi-Design Point Methodology is proposed for aeroengine conceptual design. The presented methodology adopted an unconventional approach of engaging the reliability prediction by artificial neural network (ANN) surrogate models rather than the time-consuming Monte Carlo (MC) simulation. Based on the Adaptive Particle swarm optimization, the utilization of the pre-training technique optimizes the initial network parameters to acquire better-conditioned initial network, which is sited closer to designated optimum so that contributes to the convergence property. Moreover, a new hybrid algorithm is presented to integrate the pre-training technique into neural network training procedure in order to enhance the ANN performance. The proposed methodology is applied to the cycle design of a turbofan engine with uncertainty component performance. The testing results certify that the prediction accuracy of pre-trained ANN is improved with negligible computational cost, which only spent nearly one-millionth as much time as the MC-based probabilistic analysis (0.1267 s vs. 95,262 s, for 20 testing samples). The MC simulation results substantiate that optimal cycle parameters precisely improve the engine overall performance to simultaneously reach expected reliability (≥98.9%) in multiple operating conditions without unnecessary performance redundancy, which verifies the efficiency of the presented methodology. The presented efforts provide a novel approach for aeroengine cycle design, and enrich reliability design theory as well.

**Keywords:** aircraft engine; performance reliability; cycle design; artificial neural network; pre-training technique

---

## 1. Introduction

It is a truth universally acknowledged—that an aviation vehicle in possession of a good flight performance, must be in want of a reliable propulsion system. Currently, considerable attention has been paid to the candidates of next generation aviation propulsion solution, such as variable cycle aeroengine [1], hypersonic precooled engine [2,3]. One noteworthy feature of these candidate solutions is the excellent multi-mission adaptability due to their special functions, such as variable geometry schedule, multi-cycle coupling mechanism [4], the regenerative cooling system [5], etc. Through implementing these innovative functions, the new propulsion solutions could combine the specialties of turbojet and turbofan, including the high specific thrust in supersonic penetration and low specific fuel consumption in subsonic cruise. Impressively, these specialties were supposed to be contradictory and could not simultaneously be achieved on the conventional aircraft engine.

Despite these considerable advantages mentioned above, one essential issue has not been addressed and would certainly preclude their potential application. That is the worsening overall reliability

problems. Obviously, the realization of above functions requires more complex mechanical structures, such as the extra core driven fan stage for airflow pressurization in second bypass, throttle valves for switching thermal cycle mode, and coupled heat transfer system for hypersonic propulsion [6]. As the novel aeroengine mechanical structure becomes more complex, the following overall reliability problems would undoubtedly be even worse than that of the conventional gas turbine engine.

Based on the marginal design philosophy, the solution is to maintain a certain level of redundancy by improving design performance indicators [7]. Essentially, the aviation gas turbine engine is a kind of thermal power plant, which exploits the Brayton cycle and employs air as the working medium for repeated compression, heating, expansion, and exothermic processes [8]. Through the basic thermal cycle analysis, there are two ways to boost Brayton cycle performance behavior. One is to increase the cycle pressurization ratio for higher thermal efficiency; while the other is to improve the adding heat for more output cyclic work [9]. Therefore, conclusions are conducted to enhance design performance of the gas turbine engine. Firstly, increasing the pressure ratio of the compression system could result in higher aeroengine thermal efficiency for raising the fuel-economy; secondly, raising the combustor outlet temperature could obtain more cyclic work so that aeroengine has access to more net thrust. Therefore, it is up to the corresponding key design parameters, which determined design performance indicators.

To date, the traditional deterministic single-design point methods are still widely applied to redundant performance design in aeroengine conceptual design phase. This methodology only assess single design point then evaluates other critical operating condition in the off-design phase. Thus, it is incapable to integrate of the requirements and constraints at the different operating conditions into the on-design cycle analysis, which is urgently required for future aviation propulsion schemes that noted for excellent multi-mission adaptability. Besides, the decision makers subjectively determine the increments of key design parameters by referring to the deterministic analysis results with safety factors. These situations might cause either insufficient performance redundancy in actual flight mission, or extra technical problems and increase of manufacturing cost due to potential performance waste. On the one hand, a small increase in design parameters leads to insufficient performance redundancy, which results in flight mission failure or threatens flight safety in the extreme environmental conditions. For instance, an aircraft equipped with a long-term used engine with uncertain performance degradation might be incapable to reach the desired flight range or combat radius in inclement weather conditions. Moreover, its actual taking-off running distance might exceed the safety expectation due to the insufficient net thrust in hot days. On the other hand, excessive performance redundancy results in manufacturing cost increasing and brings extra troubles for turbomachinery component design as well. The reason is that the aerodynamic and structural design of turbomachinery are strictly in accordance with the aeroengine overall performance requirements, which are substantially determined by the key design parameters. For example, the high-pressure turbine needs to enhance its cooling system or require more advanced high temperature resistant materials for blades to withstand the hotter gas flow from combustor outlet. Without new technical breakthroughs in axial compressor aerodynamic design, further boosting always means extra pressurization stages. These not only challenge the structural strength design due to the increase of weight and size, but also escalate the costs of producing a new engine and even the aircraft. To sum up, the aeroengine conceptual design is already an extremely complicated problem, which involves the interaction of each component and the coupling of multiple disciplines. When the existence of uncertainty factors cannot be ignored, solving this problem becomes more difficult. Therefore, the traditional conceptual design method is facing challenges and it is worthwhile devoting much effort to this.

To overcome the above challenge, the reliability-based multi-design point methodology is proposed for the conceptual design of aeroengine. Employing the artificial neural network (ANN) surrogate models, this unconventional approach integrates performance reliability analysis under multiple working conditions, and facilitates the optimization design procedure. The proposed methodology

could efficiently obtain the feasible design scheme, which precisely creates the expected performance redundancy and also conduces to the control of technical risk and manufacturing cost.

This paper is organized as follows. Section 2 presents a literature review. Section 3, the studies on artificial neural network for performance reliability prediction, including the generation and preprocessing of training samples and a hybrid algorithm for neural network training with the pre-training technique. In Section 4, the modeling of component-level aeroengine with uncertainty component performance is introduced, and the reliability-based multi-design point methodology is illustrated to determine the key design parameters. Section 5 validates the proposed methodology by the application of reliability-based aircraft engine thermodynamic cycle design. Conclusions and perspectives are given in Section 6.

## 2. Literature Review

The investigation of uncertainty is the basis of uncertainty-based analysis and design problem research. Currently, uncertainty is mainly classified into two types, including epistemic uncertainty and stochastic uncertainty. More specifically, epistemic uncertainty is a potential inaccuracy due to the incompleteness in knowledge (either in historical/statistical data or theories) [10,11]. Thus, a progress of knowledge or more collected data are beneficial for eliminating epistemic uncertainty. Stochastic uncertainty describes the inherent variation of the physical system or the environment under consideration, and it appears more frequently in the actual situation of aviation and aerospace engineering [7]. Emerged as the effective tools to solve stochastic uncertainty problem, probabilistic methods have attracted extensive interest. So far, probabilistic-based analysis and design methods have been successfully conducted in the fields of aerospace engineering and civil engineering that mainly focused on reliability and robustness of aerodynamic design, control system, space vehicle structure, and aeroengine alloys. For instance, PW Corp developed the Probabilistic Design System for gas turbine rotors, which integrates existing deterministic and probabilistic design techniques to assess mechanical failure modes for developing lighter weight engine components [12]. Nowadays, probabilistic analysis methods effectively promote the research advances in superalloys, which are extensively used in gas turbine engines owing to the excellent corrosion resistance and mechanical properties. The assessment of the global stability and reliability of GH4133B superalloy is implemented by using three-parameter Weibull distribution model [13]. Based on finite element simulation, the probabilistic analysis methods precisely measured the surface properties and fatigue life of Incoloy A286 alloy [14]. Lately, probabilistic methodology is successfully applied in the emerging field of aviation additive manufacturing. The newly proposed inspection scheduling approach of gas turbine welded structures considers the influences of uncertainty in material properties, weld geometry, and loads, etc. [15]. Using the first-order reliability method, the proposed routine accessed the failure rate and inspection intervals of the welded components to reduce the computational cost of probabilistic defect-propagation analysis [16]. In recent years, tremendous research has been donated to the cumulative effect of uncertainty on aeroengine overall performance. Based on Monte-Carlo probabilistic analysis method, Chen et al. proposed a feasible methodology to quantify the impacts of uncertainty in component performance on the overall performance of conventional gas turbine engine [17]. Their later research has studied the impact of component performance deviation (CPD) on adaptive cycle engine in multiple operating conditions. An interval analysis method was presented to set the standard of CPD based on the first order Taylor series expansion, which only require less computation [18]. Furthermore, this research team proposed a linear substitute model for the rapid quantification of uncertainty, which greatly simplifies the computation process and can be easily applied to other complex non-linear energy systems [19].

Above studies prove that the probabilistic method is an effective method to considerably solve the uncertainty-based problem of aeronautical science field. On this basis, the probabilistic method is introduced to solve the design problems related to the uncertainty overall performance of aeroengine. In order to assist the decision maker during the early stages of ambiguity engine design process, Mavris

proposed to utilize the probabilistic methods for analysis of the effects of component performance uncertainty on the sizing of an unmanned combat aerial vehicle engine, including payload, range, maneuver requirements [20]. Besides, the utilization of probabilistic design methods was further applied to the commercial aircraft engine preliminary design process. At first, the impacts of the uncertainty on the overall performance was quantitatively assessed, including design range, fuel burn, and engine weight [21]. What's more, probabilistic methods were exploited to analytically design the cycle parameters in the presence of uncertainty, based on the results of probabilistic sensitivities [22]. In order to solve multivariate constrained robust design problem, Mavris' team also proposed an alternate approach to probabilistic design method based on a Fast Probability Integration technique. Moreover, the following results indicated that feasible robust design solutions can be obtained and verified the efficiency of proposed method [23]. Mavris and Oliver quantitatively analyzed the influence of non-controllable parameters on aircraft reliability, and adjusted relevant flight control parameters to achieve the optimal flight performance reliability of the aircraft under different working conditions [24]. Other researches have also been conducted to acquire appropriate design schemes and prevent the adverse effects of uncertainty. To minimize technical risk, Tong et al. assessed the uncertainty impacts of novel technologies on engine overall performance, such as the fuel economy and pollutant emission, based on probabilistic analysis methods [25]. Gorla quantitatively analyzed the influences of uncertainty factors on a gas turbine power plant that operated in the wilderness, and optimized design parameters for reaching to the expected performance reliability [26]. Commonly, these studies referred to structural reliability design method and managed to settle the quantitative analysis of uncertainty. Then selection of the key design parameters is further researched to guarantee performance reliability in one particular operating condition. Above studies indicate that performance analysis and design methods are effort to evolve from certainty to probability, which can promote the design of gas turbine engine and even other non-linear energy systems.

Despite these encouraging progresses, it ought to be noted that current aeroengine performance probabilistic design research is merely limited to the most probable point (MPP) that derived from the basic thought of structural probabilistic analysis method. The essential reason for this research situation is that the implementation of probabilistic design entirely relies on Monte Carlo (MC) simulation to acquire the reliability value by calculating the probability distribution of concerned parameters. This limitation results in unaffordable computing burden to realize reliability-based multi-design point design and hampers the effort of developing the next generation aviation propulsions that concentrate on multi-task adaptability. Above all, further research is required to develop the novel conceptual design methodology to address this issue. Therefore, this paper presents the ANN surrogate models to replace the MC simulation for comprehensive reliability calculation, which significantly reduces the computational cost.

## 3. Basic Theory

The key to improving computational efficiency is to avoid performing the time-consuming probabilistic analysis by thermodynamic-based aeroengine simulation model. The critical element in the pursuit of this quest is to construct the surrogate models with acceptable precision. In this study, artificial neural network (ANN) is utilized to establish surrogate models to predict the aeroengine performance reliability under multiple flight conditions. However, through the preliminary study, it was found that the prediction of trained ANN cannot reach expected accuracy. To improve prediction accuracy of ANN, the pre-training technique is exploited to facilitate the ANN training procedure, which is implemented based on the hybridization of Adaptive Particle Swarm Optimization (APSO) and Levenberg–Marquardt Algorithm (LM) algorithm.

### 3.1. Artificial Neural Network

Inspired by biological nervous systems, ANN is a simplified computation representation that imitates the human nervous operation mechanism, and has been widely used to analyze complex

nonlinear problems [27]. ANN combines multiple nonlinear processing layers, using simple elements operating in parallel, as shown in Figure 1. It consists of an input layer, one or several hidden layers, and an output layer. The layers are interconnected via nodes, or neurons, with each hidden layer using the output of the previous layer as its input.

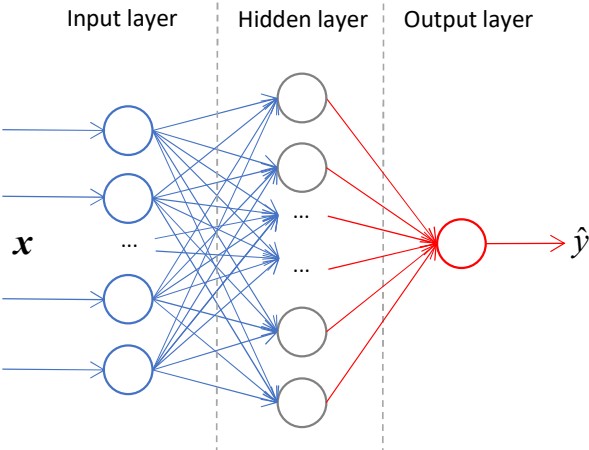

**Figure 1.** Artificial neural network.

The significant function of ANN model is to approximate the implicit non-linear relationship between input and output variables through extracting the features of the training scenarios. The ANN formulation between output response $y$ and input variables $x$ can be structured as

$$\hat{y} = f_2 \left( \sum_{j=1}^{n} w_{jk} f_1 \left( \sum_{i=1}^{m} w_{ij} x_i + b_j \right) + b_k \right) Given$$
$$\left\{ \left( x^{(1)}, y^{(1)} \right), \dots, \left( x^{(N)}, y^{(N)} \right) \right\}, want \hat{y} \approx y \tag{1}$$

where, $w$ is the connection weight; $b$ is the threshold; $f(\cdot)$ is the activation function; $m$, $n$ are the node number of input layer and hidden layer, respectively.

Neural network training is the process of feeding appropriate quantities of training samples and adjusting the network structure (mainly adjusting weights) through certain algorithms to make the network output conform to the expected value. In what follows, the procuring high-quality training samples and a new hybrid algorithm for enhancing network training are studied in detail.

*3.2. Generation and Pre-Processing for Training Scenarios*

In this study, training scenarios are generated by Latin Hypercube Sampling (LHS) method, which is capable to achieve a relatively small but a representative number of samples with different combinations of input parameters. In order to extract m samples in n-dimensional vector space, the implementation of LHS is interpreted as follows [28]:

(1) uniformly divide the cumulative distribution of each variable dimension into m non-overlapping intervals;
(2) randomly select a value from each interval;
(3) randomly combine m values of each variable are with that of other variable.

To eliminate the negative influence from the different measurement units of each input parameter, sample pre-processing technique is adopted to normalize each variable dimension to the same numerical range. Based on the Z-score method, the normalized value of samples $\{x^{(i)}, i = 1..., N\}$ in jth dimension is

$$\hat{x}_j^{(i)} = \frac{x_j^{(i)} - \mu_j}{\sigma_j} \tag{2}$$

where, $\mu$ and $\sigma$ respectively are the mean and standard deviation of samples $x^{(i)}$ in jth dimension. Thus, each element of $x^{(i)}$ such that columns of $x^{(i)}$ are centered to have mean 0 and scaled to have standard deviation 1.

The probability statistical analysis is undertaken to determine the performance reliability of each training sample. Based on the MC simulation of thermodynamic-based aeroengine model, the performance reliability output of $k$ state is presented as

$$R_k^{(i)} = \Phi\left( \frac{f_{Mean}\left(y_k\left(x^{(i)}, r\right) - y_k^*\right)}{f_{Std}\left(y_k\left(x^{(i)}, r\right) - y_k^*\right)} \right) \tag{3}$$

where, $\Phi(\cdot)$ is the standard normal distribution function; $f_{Mean}(\cdot)$ and $f_{Std}(\cdot)$ are the functions for calculating mean and standard deviation, respectively; $r$ is the vector of uncertainty parameters characterized by a predefined probability distribution; $y(\cdot)$ is the performance output response of $k$ state; $y^*$ is the required performance output. By the way, the obtained samples could be easily transformed by adjusting the value of $y^*$.

Above all, the pre-processed training samples, which contain normalized input parameters and corresponding reliability outputs in all states, are

$$\left\{ \left( \hat{x}^{(i)}, R^{(i)} \right), i = 1, \ldots, N \right\} \tag{4}$$

In general, the ANN is trained by Levenberg–Marquardt Algorithm, which started from a set of random initial network parameters according to a given prior distribution. Especially, these different initial networks parameters can converge to different local optima and may exhibit different convergence properties and network performance. Differing from a random initial network, a well pre-trained initial network is sited closer to designated optimum, which improves the convergence property and shortens the follow-up training time [29]. Thus, pre-training technique is adopted to acquire better-conditioned initial network.

To address this issue, a new hybrid algorithm of APSO and LM (HAPSOLM) is presented to integrate the implementation of pre-training and training process.

### 3.2.1. Formulation of Pre-Training

The aim of pre-training is to train the initial network to a local non-convex basin and acquire the relatively high-precision initial value, including connection weights and threshold values [30]. Thus, the pre-training process can be formulated as an optimization problem. The aim of optimization is to minimize the prediction error so that the objective function is formulated as

$$f_{err} = \|R\left(\hat{x}^{(k)}\right) - R^{*(k)}\|_2, k = 1, \ldots, M \tag{5}$$

where, $R(\cdot)$ is the network reliability prediction; k is the number of test samples; superscript * represents the actual sample reliability value.

Therefore, the optimization formulation for ANN pre-training can be presented as

$$\begin{aligned} find\ x &= \left\{ IW_i,\ b_j \right\} \\ \min & f_{err} \\ s.t \begin{cases} IW_{i,min} \leq IW_i \leq IW_{i,max} \\ b_{j,min} \leq b_j \leq b_{j,max} \end{cases} \\ i &= 1, 2, \ldots l;\ j = 1, 2, \ldots m \end{aligned} \tag{6}$$

### 3.2.2. Adaptive Particle Swarm Optimization (APSO)

The Particle Swarm Optimization (PSO) algorithm, which imitates the behavior of bird food searching, is wildly applied to neural network training, clustering analysis and the optimization [31]. In the evolutionary process of standard PSO, the velocity and position of particle i are updated as

$$v_{id}^{(k+1)} = v_{id}^{(k)} + c_1 r_1 \left( p_{id} - x_{id}^{(k)} \right) + c_2 r_2 \left( p_{gd} - x_{id}^{(k)} \right) \tag{7}$$

$$x_{id}^{(k+1)} = x_{id}^{(k)} + v_{id}^{(k+1)} \tag{8}$$

where, $c1$ and $c2$ are the acceleration coefficients; $r_1$ and $r_2$ are independent uniformly distributed random numbers in range of 0 to 1; $p_{id}$ is the position where the ith particle found the best fitness in kth generation; $p_{gd}$ is the position of the global best so far.

In order to balance the capabilities of global search and local search, the time-varying adaptive inertia weight $\omega$ is presented in APSO [32]. This weight is designed to automatically change with the objective function value of the particle. When the difference between the mean prediction error is smaller than the group minimum prediction error, $\omega$ would decrease so that the ability of local search is strengthened. On the contrary, $\omega$ would increase for leading the particles to the other probable globally optimal region and prevent them from getting trapped in the local optima.

Based on this principle of time-varying adaptive inertia weight, in this paper the $\omega$ is defined as

$$\omega = \begin{cases} \omega_{min} - \frac{f_{err} - f_{err,mean}}{f_{err,mean} - f_{err,mim}} (\omega_{max} - \omega_{min}), & f_{err} \leq f_{err,mean} \\ \omega_{max}, & f_{err} \geq f_{err,mean} \end{cases} \tag{9}$$

where, $f_{err}$ is the objective function mentioned in Section 3.2.1; subscript min and max are respectively the predefined lower and upper numerical boundary.

In order to avoid invalid iteration and guarantee the convergence, the constriction factor is also introduced and defined as [33]:

$$\xi = \frac{3}{\left| 3 - \varphi - \sqrt{\varphi^2 - 4\varphi} \right|} \tag{10}$$

where, $\varphi = c_1 + c_2, \varphi > 4$.

Above all, the velocity and position of particle i on dimension d in the evolutionary process of APSO are subsequently updated as

$$v_{id}^{(k+1)} = \omega v_{id}^{(k)} + c_1 r_1 \left( p_{id} - x_{id}^{(k)} \right) + c_2 r_2 \left( p_{gd} - x_{id}^{(k)} \right) \tag{11}$$

$$x_{id}^{(k+1)} = x_{id}^{(k)} + \xi v_{id}^{(k+1)} \tag{12}$$

### 3.2.3. Levenberg–Marquardt Algorithm (LM)

The LM algorithm is designed to approach second-order training speed without having to compute the Hessian matrix [34]. Especially, the Hessian matrix can be approximated as

$$H \approx J^T J \tag{13}$$

where, $J$ is the Jacobi matrix that contains first derivatives of the network errors with respect to the connection weights and thresholds. Back-propagation is used to calculate the Jacobi matrix, which is much less complex than computing the Hessian matrix.

Besides, the gradient can be computed as

$$g = J^T e \tag{14}$$

where, *e* is a vector of network errors.

Using the gradient and the approximation to the Hessian matrix, LM algorithm updates the variables as

$$\boldsymbol{x}^{(k+1)} = \boldsymbol{x}^{(k)} + \left[\lambda^{(k)}\boldsymbol{I} + \boldsymbol{H}_k\right]^{-1}\boldsymbol{g} \tag{15}$$

where, *I* is the identity matrix; λ is the damping factor that is adjusted during the iterative process.

When the scalar λ is small, LM is nearly the Gauss Newton method by the approximation of the Hessian matrix. On the contrary, LM almost becomes the gradient descent method with a small step size. Obviously, LM algorithm combines the advantages of Gauss Newton method and gradient descent method, which can both prevent the negative influence of ill-condition Jacobi matrix and reduce the probability of the optimization process trapping into local minimum.

To sum up, the procedure of HAPSOLM is described with the pseudo code and demonstrated in Table 1.

**Table 1.** Pseudo code of HAPSOLM.

| | **APSO for Pre-Training** |
|---|---|
| Step 1 | Randomly initialize $x_{id}^{(0)}$ and $v_{id}^{(0)}$; set *k* = 0 |
| Step 2 | Calculate the fitness of particle *i*; evaluate $p_{id}$ and $p_{gd}$ |
| Step 3 | If meet the convergence or reach the maximum number of iterations, set $x_i^{\text{best}} \rightarrow x_{LM}^{(0)}$ and go forward step 5, otherwise perform step 4 |
| Step 4 | APSO-optimize $x_{id}^{(k)} \rightarrow x_{id}^{(k+1)}$ and $v_{id}^{(k)} \rightarrow v_{id}^{(k+1)}$ for inner iterations, *k* = *k* + 1 |
| | LM for training |
| Step 5 | Initialize $x_{LM}^{(0)}$ and $\lambda^{(0)}$, set *k* = 0 |
| Step 6 | LM-optimize $x_{LM}^{(k)} \rightarrow x_{LM}^{(k+1)}$ and update $\lambda^{(k)} \rightarrow \lambda^{(k+1)}$ for inner iterations, *k* = *k* + 1; updates the values of weights and thresholds |
| Step 7 | Check the Prediction error of $x_{LM}^{(k)}$ and keep record of the best $x_{LM}^{(k)} \rightarrow x_{LM}^{best}$ |
| Step 8 | If no better $x_{LM}^{(k)}$ is observed over stop generations, output $x_{LM}^{best}$, otherwise go back to step 6. |

### *3.3. Implementation of HAPSOLM Algorithm in ANN Training*

In pre-training and training process, the implementation of HAPSOLM algorithms forms two stages, shown in Figure 2. In the first stage, APSO is used to initialize the connection weights and threshold values for acquiring better-conditioned initial network. In the second stage, LM Algorithm is adopted to minimize loss function by updating the connection weight and threshold values until meet the convergence or reach the maximum number of iterations.

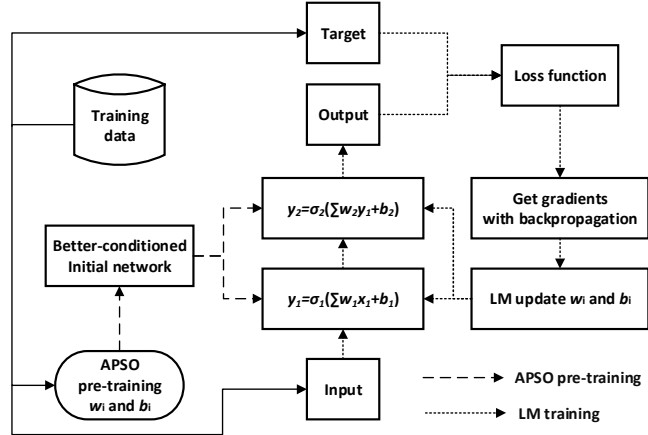

**Figure 2.** HAPSOLM in artificial neural network (ANN) pre-training and training.

## 4. Reliability-Based Multi-Design Point Methodology

The reliability-based multi-design point methodology is proposed to acquire the appropriate key design parameters by comprehensive reliability analysis for multiple operating conditions of interest. The main implementation procedure of this methodology is graphically elucidated in Figure 3 and summarized as follows:

(1)　establish the corresponding thermodynamics-based simulation model of aircraft engine with uncertainty component performance;

(2)　divide the whole flight profile into multiple critical operating conditions and simultaneously regard them as the on-design points, and then respectively determine their overall performance requirements;

(3)　generate a certain number of training scenarios and train the neural network for establishing reliability prediction surrogate models of each concerned operating condition;

(4)　construct the objective function by the attained surrogate models to facilitate the comprehensive performance reliability analysis in optimization design.

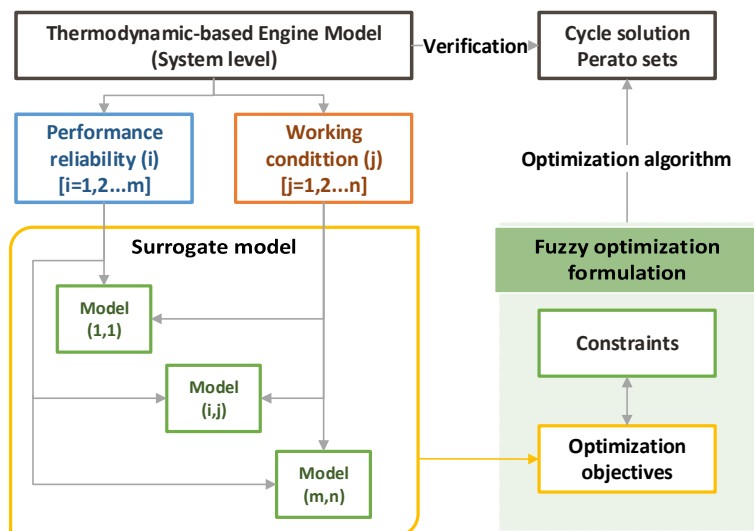

**Figure 3.** Reliability-based multi-design point approach.

Thermodynamic cycle defines the overall performance at all operating conditions, which is one of the most important attributes for a gas turbine engine [35]. Therefore, the cycle parameters are optimized as the key design parameters.

### 4.1. Aircraft Engine Modeling

In this study, we research the 2-spool mixed flow turbofan engine (MTF), which is widely equipped with air fighters. Its structure and sections are briefly shown in Figure 4.

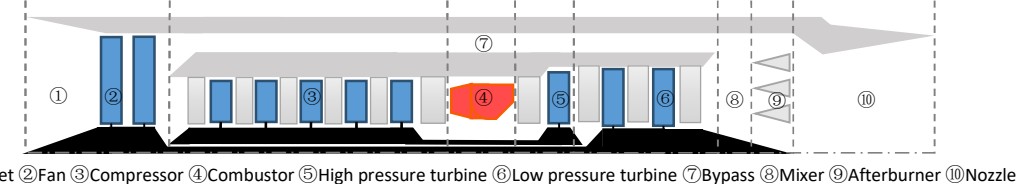

①Inlet ②Fan ③Compressor ④Combustor ⑤High pressure turbine ⑥Low pressure turbine ⑦Bypass ⑧Mixer ⑨Afterburner ⑩Nozzle

**Figure 4.** 2-spool mixed flow turbofan engine.

The aim of aircraft engine modeling is to simulate the performance behaviors of the defined thermal cycle, which is the necessary premise to obtain training samples and verify the optimal cycle solution. As shown in Figure 5, the following component-level simulation model is developed with the following characteristics:

(1)  physics based. The turbofan performance evaluation is based on principles of thermodynamics and fluid mechanics. More details about modeling technique and internal thermodynamic calculation method of each component, such as compressor, turbine, and nozzle, can be found in these references [36–38];

(2)  modular/component-based construction. Each turbofan component is developed as an individual module and communicate with each other;

(3)  incorporating engine dimensions and component maps for sizing. The component characteristic maps are physically representations of real turbo machines [39]. Through scaled within certain limits, referenced characteristic maps could simulate the performance behaviors of developed turbomachinery models.

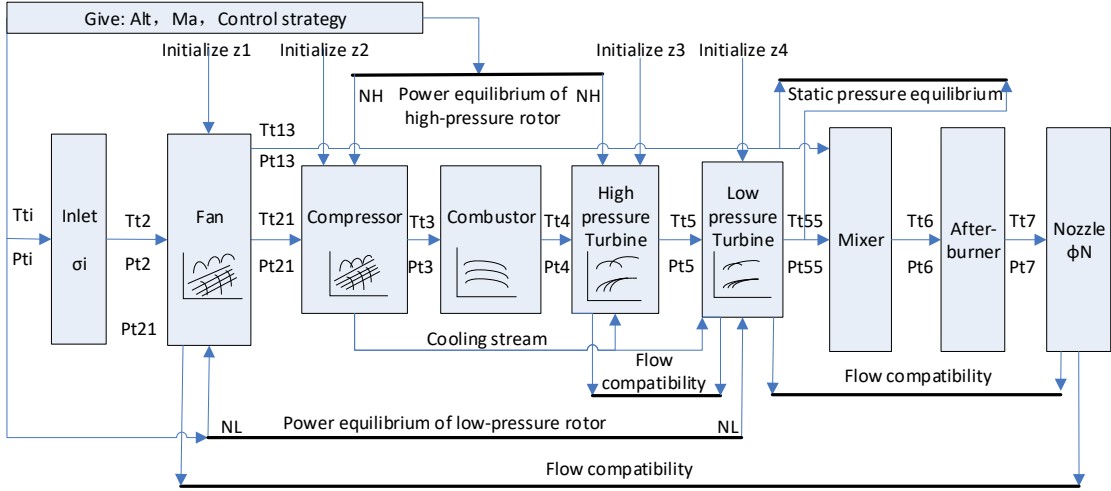

**Figure 5.** Component-level turbofan engine simulation model.

The established turbofan engine simulation model mainly consists of three modules, including the uncertainty component performance module, on-design point calculation module, and off-design point calculation module.

### 4.1.1. Uncertainty Component Performance Module

In this research, the uncertainty performance of turbomachinery components is mainly accounted to simulate uncertain effects from degradation, including fan, compressor, high-pressure turbine (HPT) and low-pressure turbine (LPT). During the long-term service, the degradation in turbomachinery components is caused by several mechanisms, such as fouling, hot corrosion and striking damage, etc. These degradation mechanisms act on the turbomachinery blade material and cause uncertain change on their shape and flow paths. Moreover, that eventually brings uncertainty effect on turbomachinery performance, including the flow capacity and efficiency, etc. [40]. To imitate uncertainty component performance, the correlative implementation method is to make the actual component performances change stochastically related to the theoretical design performance, which are defined in standard component characteristic maps. For instance, the uncertainties of mass flow and the isentropic efficiency appear in compressor during the long-term service, as displayed in Figure 6.

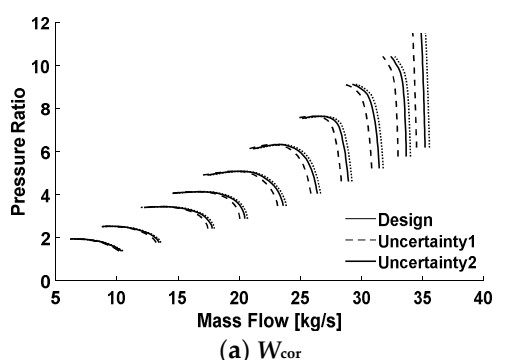
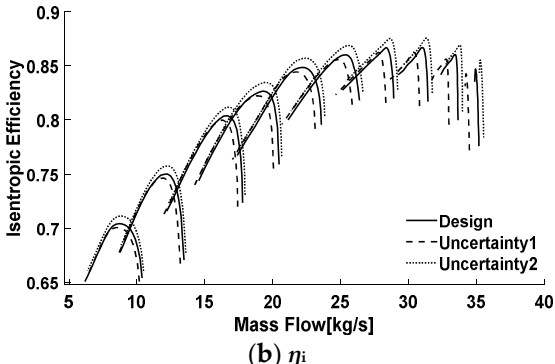

(**a**) $W_{cor}$            (**b**) $\eta_i$

**Figure 6.** The characteristic map of uncertainty compressor performance.

Random uncertainty is adopted to numerically describe the uncertainty component performances. Referring to these references [21,22], all uncertainty component performance parameters are specified as normal distribution and can be formulated as

$$\begin{bmatrix} W_{cor} \\ \eta_i \end{bmatrix} = \begin{bmatrix} r_1 C_w W_{cor,Map} \\ r_2 C_\eta \eta_{i,Map} \end{bmatrix}, r \sim N(\mu, \sigma) \tag{16}$$

where, $C$ is the map scaling factor calculated in on-design point phase; $r$ is a random number that follows a normal distribution.

Since the detailed statistical data from aircraft engine manufacturers is unavailable, the uncertainty (or noise) parameters of interest for this study are given at Table 2, according to reference [17].

**Table 2.** Component uncertain performance parameter range specification.

| Turbomachinery Component | Noise Parameter | Lower | Nominal | Upper |
|---|---|---|---|---|
| Fan, Compressor, High pressure turbine, Low pressure turbine | Mass flow | $-3\sigma$ | $\mu$ | $+3\sigma$ |
| | Isentropic efficiency | $-3\sigma$ | $\mu$ | $+3\sigma$ |

### 4.1.2. On-Design Point Calculation Module

On-design point studies compare gas turbines of different geometry, which is a necessary preparation for off-design point calculations. Many possible thermodynamic cycles and feature sizes are evaluated before a new gas turbine can be designed. In the end, a cycle is selected which constitutes the cycle design point (cycle reference point) of the gas turbine. After accomplishing the on-design point calculation, the turbomachinery component performance parameters at on-design points are determined and correlated with the relevant component characteristic maps. Some feature geometry dimensions, such as mixer outlet area and nozzle throat area, are also determined after on-design point calculation.

### 4.1.3. Off-Design Point Calculation Module

Off-design point studies the overall performance behavior of a gas turbine with selected thermal cycle and known geometry. Through off-design point simulation, the aeroengine overall performance is accessed at various operating conditions, which is specified by different flight altitude, Mach number, and control schedules.

The key to calculate gas turbine engine off-design point is to determine the collaborative working relationship of all components, which is depicted based on principles of the energy conservation and flow continuity. To determine the collaborative working relationship, 2-spool mixed flow turbofan engine requires six matching variables, which is presented as

$$z = (z_1, z_2, z_3, z_4, z_5, z_6)^T \tag{17}$$

where, $z_i$ ($i = 1,...,4$) are the auxiliary coordinates, which are respectively locating the operating point in scaled characteristic maps of each turbomachinery component; $z_i$ ($i = 5,6$) are chosen two out of the high pressure rotor relative speed, low pressure rotor relative speed and relative fuel flow, which depends on the selected control schedule.

Besides, the number of equations equals the number of variables, which guarantees each collaborative working relationship to have one and only one solution of matching variables. More specifically, six implicit nonlinear equations (or residual functions) are adopted to inspect the power balance for each rotor, the flow compatibility between connected component paths, and the airflow static pressure balance at the mixing surface boundary.

(1)　　The flow compatibility equation (residual function) of the HPT

$$(f_1(z) =)\left(W_{cor,HPT} - W'_{cor,HPT}\right)/W'_{cor,HPT} = 0 \tag{18}$$

where, $W_{cor,HPT}$ is the trial-calculation value of HPT inlet corrected gas flow; $W'_{cor,HPT}$ is the interpolation value of HPT inlet corrected gas flow referring to the HPT characteristics map.

(2)　　The flow compatibility equation (residual function) of the LPT

$$(f_2(z) =)\left(W_{cor,LPT} - W'_{cor,LPT}\right)/W'_{cor,LPT} = 0 \tag{19}$$

where, $W_{cor,LPT}$ is the trial-calculation value of LPT inlet corrected gas flow; $W'_{cor,LPT}$ is the interpolation value of LPT inlet corrected gas flow referring to the LPT characteristics map.

(3)　　The flow compatibility equation (residual function) of the aeroengine inlet and outlet

$$(f_3(z) =)\left(W_{Fan} + W_f - W_{Noz}\right)/W_{Noz} = 0 \tag{20}$$

where, $W_{Fan}$ is the fan inlet airflow; $W_f$ is the combustor fuel flow; $W_{Noz}$ is the nozzle outlet jet flow.

(4)　　The static pressure equilibrium equation (residual function) at mixer inlet

$$(f_4(z) =)\left(p_{LPT,outlet} - p_{BP,outlet}\right)/p_{LPT,outlet} = 0 \tag{21}$$

where, $p_{LPT,outlet}$ is the LPT outlet static pressure, $p_{BP,outlet}$ is the bypass outlet static pressure.

(5)　　The power equilibrium equation (residual function) of the high-pressure rotor

$$(f_5(z) =)\left(\eta_{HP}P_{HPT} - P_{Compr}\right)/P_{Compr} = 0 \tag{22}$$

where, $P_{HPT}$ is the HPT output power; $P_{Compr}$ is the compressor power consumption; $\eta_{HP}$ is the mechanical efficiency of high-pressure rotor.

(6)　　The power equilibrium equation (residual function) of the low-pressure rotor

$$(f_6(z) =)(\eta_{LP}P_{LPT} - P_{Fan})/P_{Fan} = 0 \tag{23}$$

where, $P_{LPT}$ is the LPT output power; $P_{Compr}$ is the fan power consumption; $\eta_{LP}$ is the mechanical efficiency of low-pressure rotor.

Above residual equations compose an equation group, which is

$$F(z) = (f_1(z),\ldots,f_6(z))^T \tag{24}$$

Thus, matching the collaborative working relationship is essentially solving an implicit nonlinear equation group, and can be formulated as

$$solve \ \boldsymbol{z} \ for \ \|F(\boldsymbol{z})\|_\infty \leq \varepsilon, \ \forall \varepsilon > 0 \tag{25}$$

where, $\varepsilon$ is iteration termination accuracy and usually set to $10^{-5}$.

The Newton–Raphson algorithm is adopted to approach the $z$, which is updated in each iteration [41].

$$\boldsymbol{z}^{(k+1)} = \boldsymbol{z}^{(k)} - \left(J^{(k)}\right)^{-1}F\!\left(\boldsymbol{z}^{(k)}\right) \tag{26}$$

where, $J$ is the Jacobi matrix that contains first derivatives of the residual errors with respect to the matching variables. Moreover, $J$ is formulated as

$$J = \begin{bmatrix} \frac{\partial f_1}{\partial z_1} & \cdots & \frac{\partial f_1}{\partial z_6} \\ \cdots & \cdots & \cdots \\ \frac{\partial f_6}{\partial z_1} & \cdots & \frac{\partial f_6}{\partial z_6} \end{bmatrix} \tag{27}$$

After solving z that achieves iteration termination accuracy, $z$ and flight condition parameters are re-input into the off-design point simulation model. Then all aeroengine performance parameters at the corresponding flight condition can be obtained.

### 4.2. Mission Analysis

A commonly used aircraft flight profile is graphically showed in Figure 7.

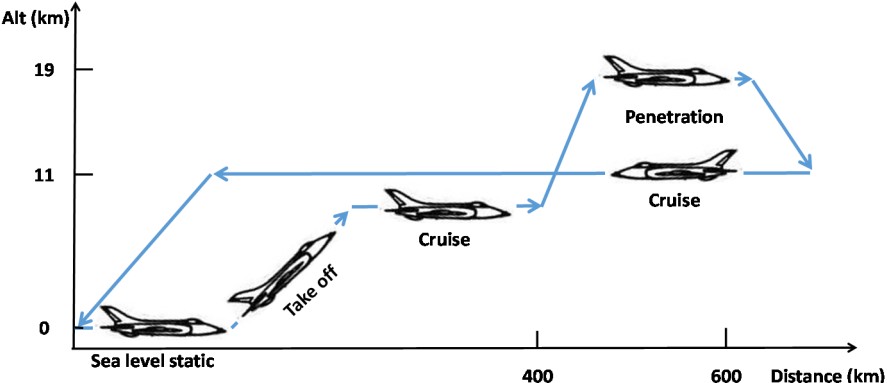

**Figure 7.** Flight mission profile.

Distributed cooperative response surface method (DCRSM) is a methodology to decompose the complex mathematical model, and then purposefully reconstruct its surrogate model in order to analyze the internal collaborative relationship more precisely and efficiently [42]. Inspired by basic thought of DCRSM, the whole flight profile is divided into four operating conditions, including sea level static (SLS), take off (TkO), subsonic cruise (SbC), and supersonic penetration (SpP). Above working conditions are simultaneously considered as the on-design points of the aircraft engine and then constructed the relevant reliability prediction surrogate models.

The overall performance parameters are mainly concerned, including net thrust ($Fn$), specific fuel consumption ($SFC$) and thrust-weight ratio ($R_{TW}$). Based on the simulation results of original cycle scheme, the specifications of all on-design points and overall performance requirements of interest are listed in Table 3.

**Table 3.** Overall performance requirements in flight profile.

| Operating Condition | Atmospheric Environment | Alt (km) | Ma | Control Schedule | Overall Performance Requirements | | |
|---|---|---|---|---|---|---|---|
| | | | | | *Fn* [kN] | *SFC* [kg/(h*kN)] | $R_{TW}$ |
| SLS | ISA | 0 | 0.00 | Fixed NL, AfB-on | / | / | ≥6.75 |
| TkO | Hot day, +15K | 0 | 0.25 | Fixed NLcor, AfB-off | ≥68.21 | / | / |
| SbC | ISA | 11 | 0.90 | Fixed NL, AfB-off | ≥18.48 | ≤92.02 | / |
| SpP | ISA | 19 | 2.00 | Fixed NH, AfB-on | ≥29.57 | / | / |

*4.3. Problem Statement*

4.3.1. Design Variables

The design variables to be optimized are the cycle parameters of SLS condition, including fan pressure ratio ($\pi_F$), compressor pressure ratio ($\pi_C$), turbine entry temperature ($T_{ET}$), bypass ratio ($R_{BP}$) and engine airflow ($W_{std}$). In order to produce reasonable thermodynamic cycle scheme, the coordinate variables are bounded as in Table 4.

**Table 4.** Cycle parameter design range.

| Cycle Parameter ($x$) | $\pi_F$ ($x_1$) | $\pi_C$ ($x_2$) | $T_{ET}$/K ($x_3$) | $R_{BP}$ ($x_4$) | $W_{std}$/(kg·s$^{-1}$) ($x_5$) |
|---|---|---|---|---|---|
| Lower | 3.2 | 9.5 | 1650 | 0.6 | 110 |
| Upper | 3.6 | 10.5 | 1750 | 1.0 | 130 |

4.3.2. Optimization Objective

The objective of the model is to maximize the overall performance reliability of each operating conditions. To deal with the correlation among multi-failure modes for comprehensive reliability calculation, the objective function which can be presented as

$$min : \sum \omega_j \left| 1 - f_{ANN}^{(j)}(\boldsymbol{x}) \right| \tag{28}$$

where, $\omega$ is the weight that defined by decision maker.

4.3.3. Constraints

Except for the constraint of the design variables, thrust-to-weight ratio ($R_{TW}$) is also employed as the constraint, which is a comprehensive index to measure the aeroengine performance ability. The formula of $R_{TW}$ can be presented as

$$R_{TW} = \frac{Fn_{SLS, AfB-On}}{g \cdot W_{Engine}} \tag{29}$$

where, $g$ is gravitational acceleration; $W_{Engine}$ is the quality of engine, which is estimated as [43]:

$$
\begin{aligned}
W_{Engine} = \quad & 0.0146T_{ET}^2 + 3.949\pi + 1516.8R_{BP}^2 - 0.045W_{std}^2 - 0.426\pi T_{ET} + 9.684R_{BP}T_{ET} \\
& + 0.0392T_{ET}W_{std} - 74.37\pi R_{BP} - 1.646\pi W_{std} - 12.87R_{BP}W_{std} - 44.21T_{ET} \\
& + 690.9\pi - 13429R_{BP} - 0.3169W_{std} + 30759.267
\end{aligned} \tag{30}
$$

where, $\pi$ is the total pressure ratio.

The $R_{TW}$ is a nonlinear implicit function of $\boldsymbol{x}$, and ought not be less than the original value. Then the constraint function of $R_{TW}$ is presented as

$$R_{TW}(\boldsymbol{x}) \geq R_{TW,Origin} \tag{31}$$

### 4.3.4. Optimization Formulation

The final solution of cycle parameters can be obtained by solving the following optimization problem:

$$
\begin{aligned}
&find\ \boldsymbol{x} = \{x_i\} \\
&\min \sum \omega_j \left| 1 - f_{ANN}^{(j)}(\boldsymbol{x}) \right| \\
&s.t \begin{cases} x_{i,min} \leq x_i \leq x_{i,max} \\ R_{TW} \geq R_{TW,Origin} \end{cases} \\
&i = 1, 2, \ldots l; j = 1, 2, \ldots m
\end{aligned}
\tag{32}
$$

## 5. Results and Discussion

### 5.1. Validation of Turbofan Model

Gasturb$^{®}$ is a widely used commercial software for evaluating the performance of the most common types of gas turbines [39]. With the same input of turbofan engine parameters, the simulation results of turbofan model are compared with those of Gasturb$^{®}$.

As shown in the Table 5, the maximal error is not beyond 3%. It indicated that the MTF model could simulate the engine performance accurately.

**Table 5.** Simulation comparison results of Gasturb$^{®}$ and turbofan model. MTF = mixed flow turbofan engine.

| Operating Condition | Performance Parameter | Gasturb | MTF Model | Error (%) |
|---|---|---|---|---|
| TkO | Fn (kN) | 69.91 | 68.21 | 2.43 |
| SbC | Fn (kN) | 19.01 | 18.48 | 2.79 |
|  | SFC (kg/h*kN) | 89.77 | 92.02 | 2.44 |
| SpP | Fn (kN) | 29.54 | 29.51 | 0.102 |

### 5.2. Validation of ANN

The effectiveness of proposed HAPSOLM and the accuracy and efficiency of pre-trained ANN are validated in this section. In this study, the total number of the training samples is 280, which exceeds 30 times the number of key design variables and considered to be sufficient to adequately search the design variables space. These training samples are employed to respectively create four ANN surrogate models for reliability prediction.

Started with the same initial values of connection weights and thresholds, ANN pre-training is carried out to compare the APSO with the standard PSO. Figure 8 displays the comparison results in terms of convergence characteristics of the standard PSO and APSO for approaching the ANN pre-training.

It demonstrated that APSO can converge to a local optimum more quickly in the early phase. Besides, APSO is more capable to jump out of the local optima and find the other potential optimal region in the evolutionary processes, owing to the time-varying adaptive inertia weight. In the first stage of HAPSOLM, a better-conditional neural network is initialized that contains relatively high-precision connection weights and thresholds.

In the second stage of HAPSOLM, we gained the final connection weights and thresholds of ANN by LM-based training, including:

(1)  ANN$_1$ for the reliability prediction of Fn under TkO condition:

$$
IW = \begin{bmatrix} -3.229 & -0.972 & 6.773 & -4.346 & -2.403 \\ -2.273 & -1.310 & 7.990 & -8.926 & 9.541 \\ 1.113 & 1.054 & 0.994 & -4.923 & -2.065 \end{bmatrix}, b_1 = \begin{bmatrix} 4.849 \\ -2.438 \\ -4.667 \end{bmatrix}, LW = \begin{bmatrix} 0.085 \\ -0.999 \\ -0.089 \end{bmatrix}^T, b_2 = [0.007]
\tag{33}
$$

(2)　ANN$_2$ for the reliability prediction of Fn under SpP condition:

$$IW = \begin{bmatrix} -0.257 & -0.980 & 1.539 & 1.482 & 0.464 \\ 6.206 & 0.044 & -4.058 & 3.008 & -10.78 \\ 0.639 & 1.306 & 2.606 & 0.650 & 3.553 \end{bmatrix}, b_1 = \begin{bmatrix} 3.962 \\ 2.493 \\ -4.693 \end{bmatrix}, LW = \begin{bmatrix} 1.096 \\ -0.992 \\ -0.003 \end{bmatrix}^T, b_2 = [-1.079] \quad (34)$$

(3)　ANN$_3$ for the reliability prediction of Fn under SbC condition:

$$IW = \begin{bmatrix} 2.334 & 1.497 & -8.591 & 9.910 & -9.617 \\ -7.756 & 0.221 & 11.08 & -6.747 & 7.102 \\ 36.63 & -41.53 & 18.26 & -17.09 & 12.48 \end{bmatrix}, b_1 = \begin{bmatrix} 2.679 \\ 10.42 \\ -29.34 \end{bmatrix}, LW = \begin{bmatrix} 0.975 \\ 0.105 \\ -0.001 \end{bmatrix}^T, b_2 = [0.084] \quad (35)$$

(4)　ANN$_4$ for the reliability prediction of SFC under SbC condition:

$$IW = \begin{bmatrix} 0.472 & 2.906 & -0.475 & 3.724 & 0.036 \\ -2.941 & -0.153 & 4.589 & 0.012 & -0.289 \\ -1.857 & -1.439 & 3.815 & -6.196 & 0.674 \end{bmatrix}, b_1 = \begin{bmatrix} -0.390 \\ 2.601 \\ -1.737 \end{bmatrix}, LW = \begin{bmatrix} 0.589 \\ 0.411 \\ -0.418 \end{bmatrix}^T, b_2 = [-0.403] \quad (36)$$

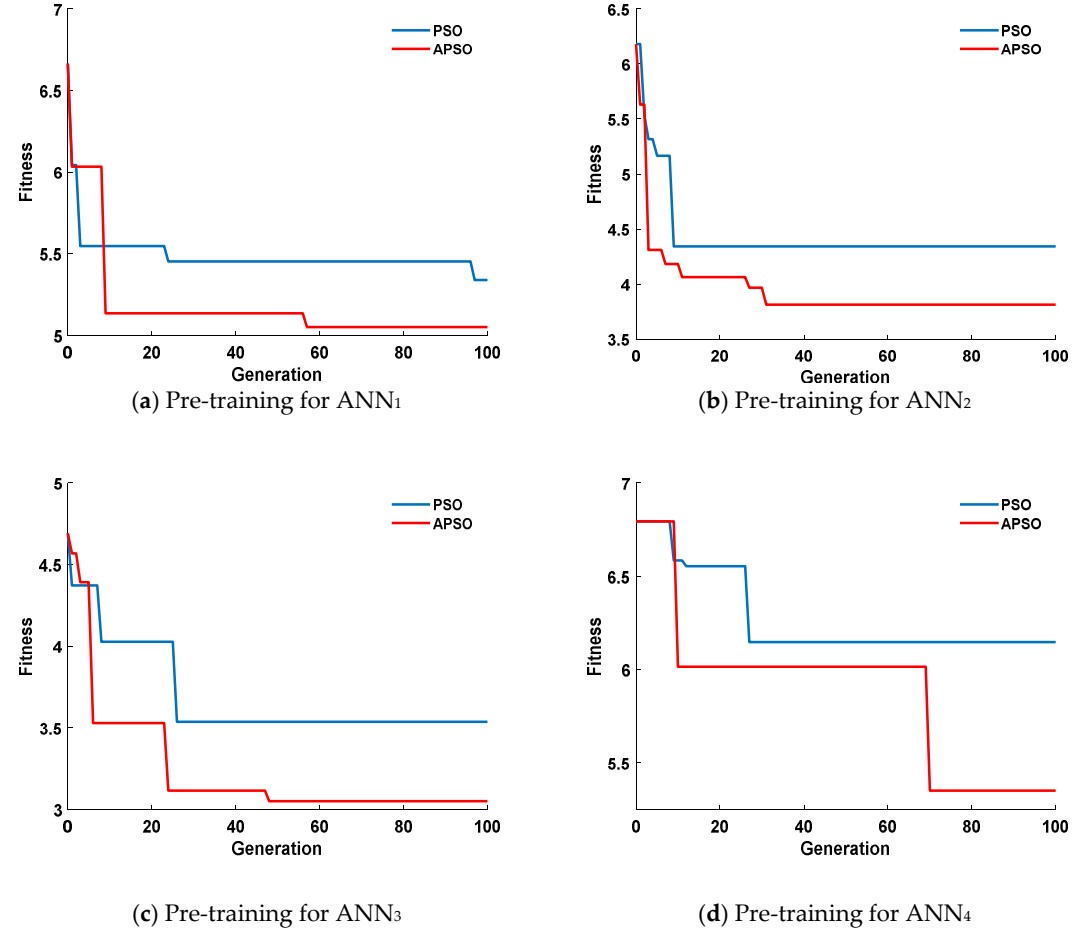

(**a**) Pre-training for ANN$_1$　　　　　　　　　　　　(**b**) Pre-training for ANN$_2$

(**c**) Pre-training for ANN$_3$　　　　　　　　　　　　(**d**) Pre-training for ANN$_4$

**Figure 8.** Pre-training convergence performance of the standard Particle Swarm Optimization (PSO) and Adaptive Particle Swarm Optimization (APSO).

Another 20 data sets were utilized to validate the effectiveness of the HAPSOLM and the MC simulation results of testing samples are regarded as the expected outputs. Orderly arranges the

samples according to reliability values from small to large; Figure 9 graphically compares the predictions of ANN and pre-trained ANN.

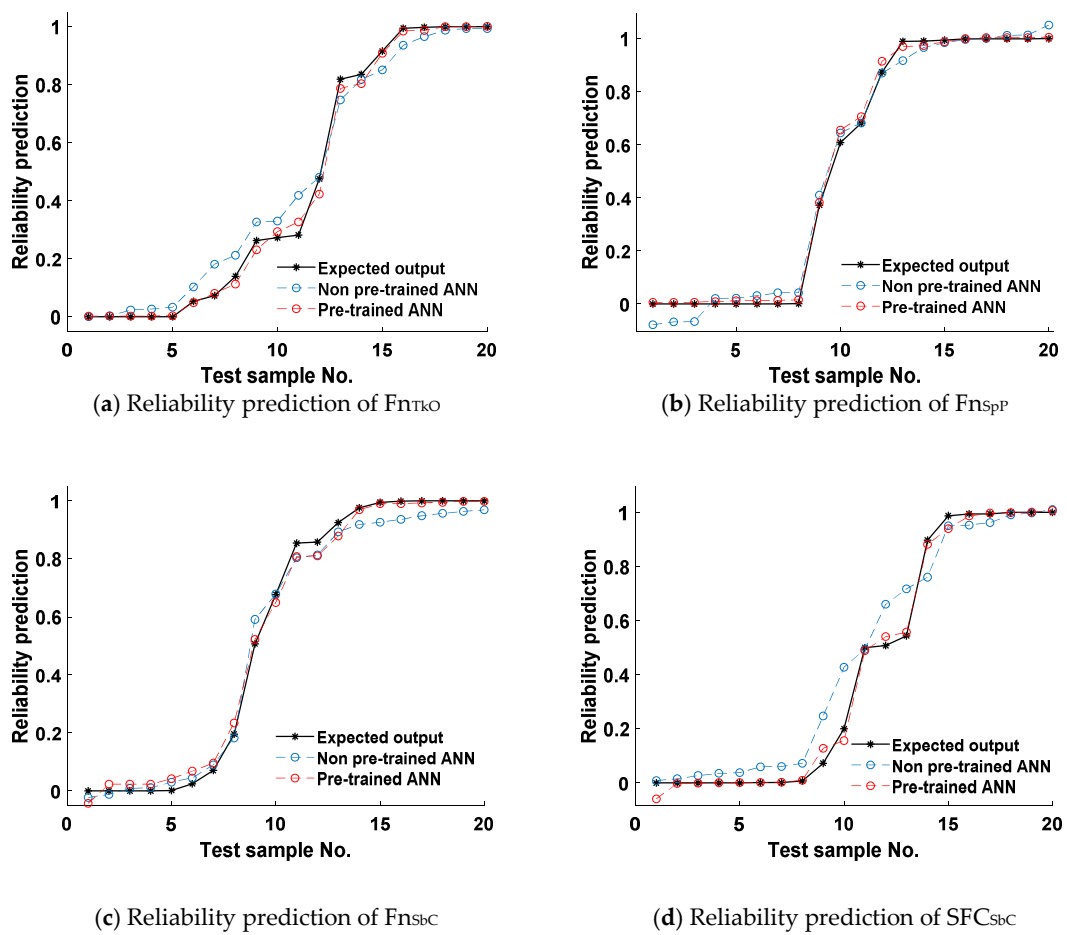

(**a**) Reliability prediction of $Fn_{TkO}$        (**b**) Reliability prediction of $Fn_{SpP}$

(**c**) Reliability prediction of $Fn_{SbC}$        (**d**) Reliability prediction of $SFC_{SbC}$

**Figure 9.** Prediction comparison of the ANN and pre-trained ANN.

It is seen that the prediction results of ANN and pre-trained ANN are basically consistent with MC results. However, there are some obvious differences between the results of ANN and MC in some regions, but the results of pre-trained ANN are highly consistent with those of MC in all regions. The results demonstrate that the prediction of pre-trained ANN is more reasonably matching the expected output values, compared with the ANN prediction.

Above simulations are carried out on the same machine consisting of an Intel Core 2.8-GHz processor and 8-GB DDR3 memory. Computing time, absolute error of the target value, and predicted value are employed to further measure the computational efficiency and accuracy, as illustrated in Table 6.

The computational time of ANN and pre-trained ANN is approximately 0.13 s, which is far less than that (95,262.2 s) of MC simulation based on thermodynamic-based aeroengine model. Moreover, the maximal absolute error of pre-trained ANN is not beyond 0.06 and the average absolute error is not more than 0.03, which is obviously more accurate than ANN (0.228 and 0.0657). The above results verify the efficiency and accuracy of pre-trained ANN within the range of the training data covered.

**Table 6.** Comparison of computing time and precision (20 testing samples). MC = Monte Carlo.

|  |  | MC ($10^3$ times) | ANN | Pre-trained ANN |
|---|---|---|---|---|
| **Computing Time (s)** |  | **95,262.2** | **0.1274** | **0.1267** |
| Averageabsolute error | $R_{Fn,TKO}$ | \ | 0.0426 | 0.0149 |
|  | $R_{Fn,SuA}$ | \ | 0.0319 | 0.0134 |
|  | $R_{Fn,SuC}$ | \ | 0.0394 | 0.0254 |
|  | $R_{SFC,SuC}$ | \ | 0.0657 | 0.0173 |
| Maximumabsolute error | $R_{Fn,TKO}$ | \ | 0.0137 | 0.0542 |
|  | $R_{Fn,SuA}$ | \ | 0.0782 | 0.0471 |
|  | $R_{Fn,SuC}$ | \ | 0.109 | 0.0475 |
|  | $R_{SFC,SuC}$ | \ | 0.228 | 0.0592 |

*5.3. Optimization Solution*

APSO is also exploited to solve the optimal value of cycle parameters. Figure 10 plots the global best value at each iteration. The observation is that the fitness value is gradually stable after 60 generations through four drops.

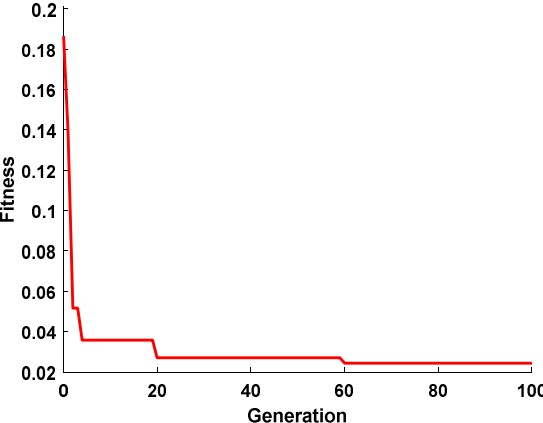

**Figure 10.** Fitness curve in the APSO evolutionary processes.

According to the result of MC simulation computing time mentioned above, it would cost approximately 4760 s for one particle to calculate its fitness value that is associated with reliability in four states. In each iteration of the optimization process, it is not difficult to estimate that computation time is at least 39.6 h if the practice population size is 30. So, it would spend nearly 165.3 days to complete one optimization calculation when the evolutionary generation is 100, which is totally unaffordable. This situation reveals the vital issue that MC-based probabilistic design is extremely difficult to implement so that only concentrates on MPP and ignores other working conditions. Nevertheless, it only spent 224.2 s to complete one optimization procedure based on the reliability prediction of ANN surrogate models, which dramatically improved the calculation efficiency with acceptable accuracy.

The optimal solution of the design problem is summarized in Table 7.

**Table 7.** The optimal solution of cycle parameters.

| | **Cycle Parameters** | | | | | | |
|---|---|---|---|---|---|---|---|
| | $\pi_F$ | $\pi_C$ | $T_{ET}$ (K) | $R_{BP}$ | $W_{std}$(kg/s) | $R_{TW}$ | Fitness |
| Origin | 3.302 | 9.750 | 1715 | 0.7900 | 122.7 | 6.75 | / |
| Optimized | 3.525 | 10.305 | 1727.2 | 0.8484 | 129.8 | 7.42 | 0.0243 |

According to design boundary of cycle parameters and $R_{TW}$ requirement, the APSO is certainly functioning as expected in terms of rejecting unfeasible solutions that violate constraints. Compared with the original cycle scheme, the cycle parameter values of optimal solution are increased in varying levels, which is consistent with the expectation of increasing performance redundancy.

In MC verification process, we input the candidate cycle solution to MTF model and then set sample capacity of 2000 for all operating condition. Figure 11 visually compare the overall performance frequency histogram of candidate solution and the original cycle scheme.

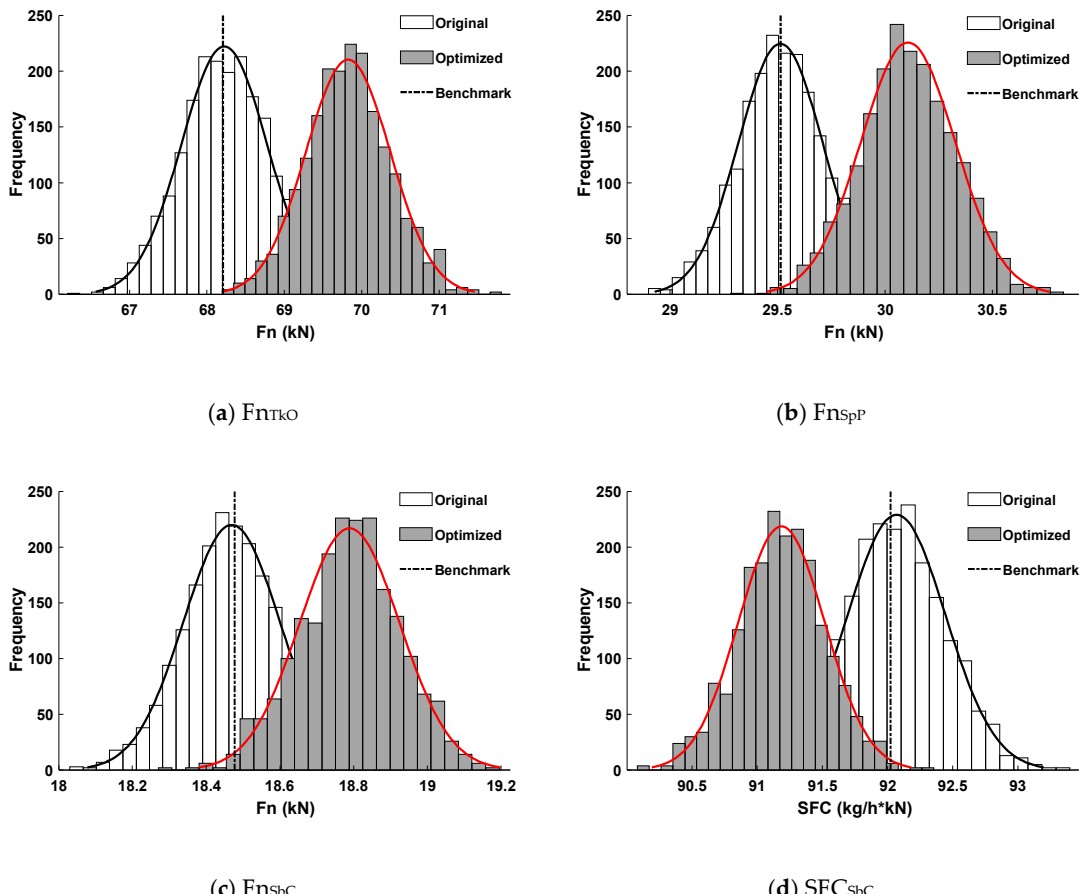

(**a**) $Fn_{TkO}$

(**b**) $Fn_{SpP}$

(**c**) $Fn_{SbC}$

(**d**) $SFC_{SbC}$

**Figure 11.** Aeroengine overall performance frequency histogram.

The first observation is that the performance output responses approximately obey normal distribution. Secondly, the candidate cycle solution significantly shifts the performance distribution in the direction of improved performance reliability, but not far away from the benchmark lines of all operating conditions of interest.

Referring to the Formula (4), further statistics are carried out and the results are displayed in Table 8.

**Table 8.** Statistics results of candidate cycle solution.

|  | TkO | SpP | SuC | |
| --- | --- | --- | --- | --- |
|  | $R_{Fn}(\%)$ | $R_{Fn}(\%)$ | $R_{Fn}(\%)$ | $R_{SFC}(\%)$ |
| Origin | 52.48 | 53.53 | 47.91 | 46.26 |
| Optimized | 99.85 | 99.66 | 98.90 | 99.42 |

The results demonstrated that optimized solution make all target performance reliability not less than 98%, which make a significant improvement over the original cycle scheme. For instance,

the 95% confidence interval of the specific fuel consumption ([90.51,91.84]kg/(s*kN)) moves bodily below the corresponding benchmark (92.03 kg/(s*kN)). Therefore, this has resulted in a significant improvement for aeroengine fuel-economy. Thus, this considerably enhance the aircraft capability of maintaining flight range with the cumulative effect of various uncertainties. In addition, the values of all target performance reliability do not exceed 100%, which manifest that the increments of key design parameters are rational without potential performance waste. Hence, the candidate cycle solution is feasible according to the reasonable performance redundancy, which would nearly yield the best possible product with minimal risk and the minimal increase in manufacturing costs.

## 6. Conclusions

This paper has presented reliability-based multi-design point methodology for a preliminary design oriented to the next generation aviation engine. The necessary mathematical tools of presented methodology are elucidated in detail. The proposed design method is applied on the cycle design of the MTF engine model with uncertainty component performance, and the following results are visualized and quantitative discussed. The main conclusions are drawn as follows:

1. the presented hybrid algorithm integrates the APSO-based pre-training technique into the network training procedure. It respectively reduced the average error and maximum error of ANN prediction at least 1.4% and 3.1%, which enhanced the performance of ANN;
2. the utilization of the ANN surrogate models facilitate the reliability-based cycle design optimization, which replaces the time-consuming probabilistic analysis based on MC simulation (about 4760 s) and only requires negligible computing time cost (about 0.13 s) for comprehensive reliability prediction;
3. the optimization design solution of presented methodology reasonably increases the aeroengine performance redundancy to precisely reach the expected reliability of all concerned operating conditions. The optimal cycle enables the aeroengine to operate efficiently and reliably in multiple working conditions, which satisfies the critical demand of multi-task adaptability for the next generation aeroengine so that verifies the efficiency of proposed methodology;
4. the effort of this paper is to explore novel conceptual design methodology oriented to the next generation aviation propulsion solution. This methodology is universal and can be easily applied to other types of gas turbine engine.

The utilization of ANN surrogate models avoid the problem of heavy computational burdens that inherently exist in MC simulation. Thus, it facilitates the application of the reliability-based multi-design point method for candidates of future aviation propulsion, which are noted for excellent multi-mission adaptability. Based on ANN surrogate models, the presented methodology could effectively and efficiently obtain the reasonable design scheme referred to the performance requirements. This methodology addresses the limitation of traditional deterministic design method that determines the key design parameters by subjectively setting the performance redundancy. Therefore, the overall performance redundancy could be set at a reasonable level so that contributes to the technical risk management and cost control of aeroengine manufacturing. Inspired by this study, the following issues must be addressed for further applications and the effectiveness of proposed methodology:

1. other advanced deep learning techniques (deep belief network, deep reinforcement learning, etc.) should be further researched to accommodate development of high-precision surrogate model;
2. further investigation on the reliability-based design methodology is needed to apply to the integrated design of aircraft and aviation engine for design technique progress.

**Author Contributions:** Conceptualization, D.C. and G.B.; methodology, D.C.; formal analysis, D.C.; investigation, D.C.; writing—original draft preparation, D.C.; writing—review and editing, G.B. All authors have read and agreed to the published version of the manuscript.

**Funding:** This research is supported by National Nature Science Foundation of China (NSFC) under Grants 51776010, 51975028, and Project MIIT. The author is thankful for the support from Collaborative Innovation Center of Advanced Aeroengine.

**Conflicts of Interest:** The authors declare no conflicts of interest.

## Nomenclature

| | |
|---|---|
| Alt | altitude (kM) |
| $b$ | threshold of a neuron |
| $C$ | scaling factor of component characteristic map |
| $c$ | acceleration coefficients |
| $\boldsymbol{e}$ | vector of errors |
| $Fn$ | net thrust (kN) |
| $g$ | gradient |
| ISA | international standard atmosphere |
| $\boldsymbol{I}$ | identity matrix |
| $\boldsymbol{J}$ | Jacobi matrix |
| N | relative rotating speed of spool (%) |
| $P$ | power (kW) |
| $Pt$ | total pressure (kPa) |
| $p$ | static pressure (kPa) |
| $p_{id}$ | position of the individual best |
| $p_{gd}$ | position of the global best |
| $R_{BP}$ | bypass ratio |
| $R_{TW}$ | thrust-weight ratio |
| $\boldsymbol{r}$ | Vector of random number |
| $r$ | random number |
| SFC | specific fuel consumption (kg/(h*kN)) |
| $T_{ET}$ | turbine entry temperature (K) |
| $Tt$ | total temperature (K) |
| $v_{id}$ | velocity of the individual particle |
| $W$ | mass flow |
| $w$ | objective weight |
| $\sigma$ | standard deviation |
| $\eta$ | isentropic efficiency |
| $\lambda$ | damping factor |
| $\mu$ | mean value |
| $\pi$ | pressure ratio |
| $\xi$ | constriction factor |
| $\omega$ | time-varying adaptive inertia |
| *Subscript* | |
| cor | corrected parameter |
| H | high pressure rotor |
| L | low pressure rotor |
| max | predefined upper numerical boundary |
| min | predefined lower numerical boundary |
| std | standard state |

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
