# Peer review of "A Study on Aeroengine Conceptual Design Considering Multi-Mission Performance Reliability"

_applsci, doi:10.3390/app10134668_

Round 1

Reviewer 1 Report

General Comments:

The work presents a method to perform reliability-based multi-point design by employing ANNs to perform the reliability predictions instead of MC simulations. They propose a new hybrid algorithm that integrates a pre-training technique (not clear what they mean) with the actual training technique to enhance the predictive capacity of the ANN model.

On reading the technical approach, the paper seems to show a good application of the method. However, the grammar, writing style, presentation, and some technical content require a major overhaul in terms of the following:

Abstract:

  1. The first sentence in the abstract must be reworded. It does not read well as it stands in the current manuscript.
  2. Please mention how your pre-training achieves the desired effect. It is not sufficient to mention what it is intended for.
  3. The general flow is very convoluted. Please re-write with a coherent flow of thought.
  4. The use of English also needs a lot of attention. There are several grammatical mistakes and ready poorly in general. Please correct all the grammatical mistakes and re-write.

Introduction:

  1. The introduction contains all the relevant information but the organization and presentation confuse the reader instead of clearly presenting a logical progression of past work to identify a gap in how these issues are tackled today, what specific problem in the identified gap is being addressed by this work, and what is the approach being taken to achieve the goal.
  2. Extensive grammar correction and language edits are needed. Please stick to the use of professional language. For instance, in Page 3, line 112, “…But there is a fly in the ointment…” reads very colloquial and unprofessional. Please change it to something more professional. I would suggest removing it all together.
  3. Please re-write the entire section incorporating the comment 1 and 2. Reading the introduction should make the reader clear about the need for the novel approach in the grand context of the topic area.
  4. It may not be a bad idea to split the introduction into “introduction” and “background/literature review”.

Basic Theory:

  1. How training data is being generated has not been clearly explained in section 2.2. Please introduce the notation properly. In fact it is not a bad idea to have a section that purely introduces notation for the entire paper. It will greatly reduce confusion.
  2. The content is there but presentation can be greatly improved with grammar correction, proper notation definitions, and correcting the general organization of flow of thoughts.
  3. Figure 2 is misleading because it appears to show that the pre-training does require the training data. It does though, right? You are essentially trying to locate a good valley using APSO for the later training procedure optimization. Please correct this figure to clarify what the pre-training optimization achieves and explicitly show that it also requires the training data.
  4. Please re-write with grammar corrected and sentences re-organized with proper flow of thought. Define notations wherever necessary either in text or upfront in a separate section.

Reliability-based Multi-design Point Methodology:

  1. This section probably does the most decent job of explaining what is actually being done.
  2. Figures 3 can definitely use larger fonts to make it clear. Reading it is hard as it stands in the current manuscript.
  3. Figure 5 must be re-created with larger font. Please export it as a vector graphic for clarity to increase the resolution of the raster image.
  4. Figure 6 needs to have larger fonts for the labels and tick sizes for the axes.
  5. Again in subsections 3.1. and 3.2. the notation is confusing and missing in several places. Makes it unnecessarily difficult to follow. Please correct this.
  6. Again, the write-up although well-organized thought-wise still contains several grammatical mistakes and use of unprofessional language. Please re-write.
  7. All the figures need to be regenerated with either a higher resolution or in vector graphics format.

Results and Discussion:

  1. Again, the write-up although well-organized thought-wise still contains several grammatical mistakes and use of unprofessional language. Please re-write.
  2. Several notations have not been defined. Please organize the notation definitions throughout the manuscript.
  3. All the figures need to be regenerated with either a higher resolution or in vector graphics format.

Conclusions:

  1. This section summarizes the findings well. However, the grammar is poor throughout. Please re-write from scratch.
  2. Add some sentences to place the novel findings of your work back in the grand context of things you discuss in the introduction section. Place this before the future work discussion.

Author Response

The co-author and I would like to thank you for the time and effort spent in reviewing the manuscript (ID: applsci- 838137). Those comments are all valuable and helpful for revising and improving our paper, as well as the important guiding significance to our research team. We have studied all the comments carefully and have made correction. Each comment will be directly addressed regarding the modified with annotations and changes highlighted in green (shown in Microsoft Office Word 2019). The details of revision illustration can be found in a pdf file. (List of response for reviewer 1).

Reviewer 2 Report

Dear editor,

In the manuscript received, the authors develop a new way of pre-training artificial neural networks in order to obtain an accurate method than Monte Carlos with standard APSO. This work is explained from a scientific point of view correctly. The method stated by them is promising and can be transferred in medium term to the aeronautical industry. Therefore, this manuscript should be accepted with minor revisions. In particular, authors should be taken into account 3 small issues before the final acceptance:

  • In the introduction would be interesting in line 76 when they talk about the new advanced alloys, introduce any example of these alloys… (Eg: Nowadays, Inconel 718 is used but the market are researching others than Incoloy, Nimonic, etc..)
  • In section 3, line 288. Please use impersonal format, not “we choose….”
  • Regarding results section. Gasturb is a trademark, please use ®. Also would be interesting in line 460 indicating % of difference in the critical points.

Author Response

The co-author and I would like to thank you for the time and effort spent in reviewing the manuscript (ID: applsci- 838137). Those comments are all valuable and helpful for revising and improving our paper, as well as the important guiding significance to our research team. We have studied all the comments carefully and have made correction. Each comment will be directly addressed regarding the modified with annotations and changes highlighted in green (shown in Microsoft Office Word 2019). The details of revision illustration can be found in a pdf file. (List of response for reviewer 2).

Round 2

Reviewer 1 Report

The authors have made most of the requested corrections. The manuscript reads better than before now.

This manuscript is a resubmission of an earlier submission. The following is a list of the peer review reports and author responses from that submission.

Round 1

Reviewer 1 Report

Review report for applsci-795221, “A Study on Aeroengine Conceptual Design Considering Multi-Mission Performance Reliability”

In general, both “multiple mission point design” and “reliability design” are very important to engine design and optimization problems. The authors use artificial neural network (ANN) methods to build up the surrogates to further predict the engine performance. The research topic is interesting and also very important. In the following please find my detailed comments.

  1. In the abstract and introduction section, there are quite some abbreviations not explained when they first appeared in the text.
  2. In section 2 “basic theory’ the authors spent a lot of words describing the general ANN, the generation of training scenarios, etc. However, I did not see clearly how the algorithms were implemented.
  3. In section 3, the authors have shown the general methods. But the methods are too general to show how they are exactly modelled and integrated. To be specific, with the information provided in the manuscript, others could not reproduce anything. Especially “thermodynamic-based aeroengine simulation model” - I think they should be the core of the paper and cannot be neglected. For example, what about the conditions of your engine model? Are they universally applicable for different flight altitudes and speeds? What are the modelling parameters? Have you verified and validated your engine models? If the engine simulation models are not valid, the further reliability, sensitivity studies would not be meaningful.
  4. In the result section, the authors show quite promising results with the optimized cycle parameters listed in table 6. As comparison, have the authors tried some gradient based optimization methods at least for single point optimization?
  5. There are some language issues and typos that should be revised before publication.

Reviewer 2 Report

This submission is focused on a reliability-based multi-design point methodology to capture the appropriate key design parameters of a turbofan engine by comprehensive reliability analysis for multiple operating conditions. Traditionally, the engine’s designer selects a design point (conditions where the engine operates at most of time, e.g. cruise condition for an aircraft) to determine the engine parameters. Other conditions are referred to off-design performance.  In this submission, the authors divide the flight profile into multiple operating conditions with multiple design points and attempt to design engine components for best performance during all these conditions.  This reviewer feels that there is no benefit of the approach used in this paper. There are many engine design studies focused on one single design point and to improve the performance during off-design conditions. The existing design methodology is still reasonable and reliable and I am wondering how the method of this paper works better than available approaches.  Additionally, I feel that the problem statement and the methods used are not clearly detailed. Unfortunately, I cannot recommend this submission for publication mainly it fails to show an improved design method for jet engines.  

More comments:

Abstract Line 11, rephrase the sentence “However, it would also face …” The text is not clear.  Are you considering the design of a new aero engine with multiple design points compared with conventional gas turbines with one single design point?  How the new design affects the engine reliability?

Define abbreviations used through text, e.g ANN, MC, and etc.

Add a list of symbols used;  some symbols are not defined where they have been introduced.

Line 19 “to [the] cycle design [of] a turbofan engine”

Line 24: What the symbol ≮ denotes?

Line 25:  “which verifies” instead of “which verify”

Introduction:  I would suggest rephrasing the second paragraph

Line 40 “could not simultaneously [being achieved]”

Suggesting to rephrase Lines 42-49

Line 47 “structures become” delete “s”

Line 71 “conditions [or] its actual taking-off”

Line 94 Line 326 Awkward citation errors “Error! Bookmark not defined” 

Line 109, add a reference for “Mavris has pioneered”

Line 120, rephrase “Mavris has pioneered “And the following results indicated”

Line 128 “Rama S.R.” ?  Reference 21 used for this name has no author named Rama!

Avoid using “we”

Sections 2.1 to 2.3 describe too much details (including equations) of how ANN works; not needed for an article such as; briefly describe the ANN structure and training method used.

Line 304 rephrase “By the way, the aim of this paper is exactly”

Eqs 19-24:  Why a parentheses was used in the left side of these equations?
